# Developmental Acquisition of *p53* Functions

**DOI:** 10.3390/genes12111675

**Published:** 2021-10-23

**Authors:** Sushil K. Jaiswal, Sonam Raj, Melvin L. DePamphilis

**Affiliations:** 1National Institute of Child Health and Human Development, Bethesda, MD 20892, USA; sushil.jaiswal@nih.gov; 2National Human Genome Research Institute, Bethesda, MD 20892, USA; 3National Cancer Institute, Bethesda, MD 20892, USA; sonam.raj@nih.gov

**Keywords:** pluripotent, embryo, stem cells, genomic stability, cell cycle, apoptosis, differentiation, cancer

## Abstract

Remarkably, the *p53* transcription factor, referred to as “the guardian of the genome”, is not essential for mammalian development. Moreover, efforts to identify *p53*-dependent developmental events have produced contradictory conclusions. Given the importance of pluripotent stem cells as models of mammalian development, and their applications in regenerative medicine and disease, resolving these conflicts is essential. Here we attempt to reconcile disparate data into justifiable conclusions predicated on reports that *p53*-dependent transcription is first detected in late mouse blastocysts, that *p53* activity first becomes potentially lethal during gastrulation, and that apoptosis does not depend on *p53*. Furthermore, *p53* does not regulate expression of genes required for pluripotency in embryonic stem cells (ESCs); it contributes to ESC genomic stability and differentiation. Depending on conditions, *p53* accelerates initiation of apoptosis in ESCs in response to DNA damage, but cell cycle arrest as well as the rate and extent of apoptosis in ESCs are *p53*-independent. In embryonic fibroblasts, *p53* induces cell cycle arrest to allow repair of DNA damage, and cell senescence to prevent proliferation of cells with extensive damage.

## 1. Current Consensus on *p53*

*p53* is referred to as “the guardian of the genome”, because it maintains genomic stability by arresting proliferation of cells with damaged DNA, thus minimizing the risk of tumor development by maintaining a pool of healthy cells [1]. Nevertheless, *p53* is not essential for embryonic development. Mice nullizygous for *p53* (*p53*-/- mice) do not die until they succumb to tumors as adults [2,3,4]. Moreover, the *p63* and *p73* homologs of *p53* also are not essential for development, and they are not functionally redundant with *p53* [5]. Therefore, the *p53* family of proteins is not essential for mammalian development.

In differentiated cells, *p53* operates primarily through transcriptional activation of the cyclin-dependent kinase inhibitor *p21* to prevent cells from entering S-phase, and the pro-apoptotic proteins PUMA and NOXA/PMAIP1 [6,7] to activate the rapid induction of apoptosis when the effects of cell stress are too great to be reversed. Remarkably, neither *p21* nor *PUMA* are required for mouse development. *p21*-/- mice develop spontaneous tumors at an average age of 16 months, whereas wild-type mice are tumor-free beyond 2 years of age [8]. *Puma*-/- mice recapitulate virtually all of the apoptotic deficiencies in *p53*-/- mice [9].

Over half of all human tumors express a mutated *p53* in which the mutation down-regulates *p53* activity. Inactivation of the *p53* gene leads to three hallmarks of cancer: genomic instability, metastasis, and drug-resistance [10]. Restoration of wild-type *p53* protein can result in tumor regression and prolonged mouse survival [11]. Besides mutations in the *p53* gene, loss of *p53* activity can also result from epigenetic silencing, from disruption of pathways that activate *p53* in response to DNA damage, from viral proteins that impair *p53* activity, and from up-regulation of MDM2 expression [12,13,14,15,16,17]. 

In response to cellular stress such as inhibition of DNA replication or mitosis, double-strand DNA breaks, changes in the oxidative state, ribonucleotide depletion, or oncogene activation [18], *p53* activates transcription of genes involved in many different events. These include maintaining genomic stability by inducing cell cycle arrest, apoptosis or senescence [19,20,21], regulating homeostasis and metabolism [22], regulating pluripotency and differentiation [23,24], triggering inflammation [25], reducing reproduction [26], regulating aging [27], and stimulating tissue regeneration [28]. Recent evidence indicates that *p53* also has a role in enabling cells to adjust their metabolism in response to changes in glucose and other nutrient levels, oxygen availability, and reactive oxygen species [29,30]. Consequently, *p53* has been reported to induce expression of hundreds of different genes that affect viability [31].

Analysis of 3661 *p53* target genes gleaned from multiple sources concluded that only 343 are *p53* target genes with high-confidence [32]. This represents an 11-fold reduction in the number of *p53* target genes reported in the literature and suggests that *p53* targets about 1.6% of the approximately 21,000 human protein coding genes [33]. These high-confidence *p53* target genes function in multiple processes (Figure 1). 

The nature of the phenotypic response to *p53* activation is generally proportionate to the amplitude, duration, and nature of the activating signal [18]. Severe stress induces apoptosis and senescence, whereas milder stress leads to a transient growth arrest coupled with an attempt to deal with the cause of stress and repair the damage caused by it.

## 2. Current Consensus on Preimplantation Mouse Development

Mammalian development begins when an egg is fertilized by a sperm to produce a 1-cell embryo termed the zygote [34,35]. The zygote then undergoes a series of cell cleavage events to produce a blastocyst that implants into the uterine endothelium to produce an embryo (Figure 2). The 1-cell to 8-cell embryos consist of totipotent blastomeres encapsulated by a thick transparent membrane termed the zona pellucida. Totipotent cells can develop into a new organism; they can give rise to both the placenta and the embryo. Pluripotent cells can become any type of cell in the body, as exemplified by embryonic stem cells (ESCs). Multipotent stem cells can differentiate into two or more distinct cell types.

During the 8-cell to 32-cell stage of development, the blastomeres develop cell-to-cell adhesion, and the outer blastomeres differentiate into the multipotent trophectoderm while the remaining blastomeres form the pluripotent inner cell mass. The epithelial trophoblast cells (trophectoderm) give rise only to cells required for implantation and placentation, whereas the inner cell mass of the blastocyst (recognized upon formation of a blastocoel cavity) differentiates into the pluripotent epiblast and the multipotent primitive endoderm. Following implantation of the blastocyst into the uterine membrane, the primitive endoderm differentiates into multipotent visceral and parietal endoderm that give rise to internal layers of various organs. The other two multipotent primitive cell layers are derived from the epiblast. Primitive mesoderm gives rise to muscle and blood cells, whereas primitive ectoderm gives rise to skin and neuronal cells.

### 2.1. Transcription of the p53 Gene during Embryonic Development

*p53* transcripts have been detected at low levels throughout mammalian development. In mice, transcripts from genes involved in programmed cell death (*Tr**p53*, *Pdcd4*, *Bad*, *Bax*, *Bcl2l1*, and *Bcl2*) are present from the 1-cell embryo through the blastocyst stage with greatest number of *p53* transcripts/embryo in blastocysts [37]. Later studies in humans quantified the number of *TP53*, *BRCA1*, *BRCA2*, *ATM*, *RB1*, *MAD2*, *BUB1*, *APC* and *ACTB* transcripts/embryo/cell from oocytes through blastocysts [38]. The average number of *p53* transcripts/embryo was 141 in oocytes, 51 in 2–4 cell embryos, 72 in 5–10 cell embryos, and 1120 in blastocysts; the number of *p53* transcripts/cell was 21, 14, 8 and 14, respectively. Thus, the number of *p53* transcripts per cell is comparatively constant during preimplantation development. 

From zygote to adult, genes involved in cell proliferation and programmed cell death are expressed at varying levels in virtually every cell [39]. All cells express *p53* transcripts until 10 days post-coitum, but at later stages in development, only specific differentiating tissues showed high levels of expression. Terminally differentiated tissues exhibited very low levels of *p53* mRNA, suggesting a strong correlation between *p53* levels and the fraction of embryonic cells undergoing proliferation or differentiation. 

### 2.2. p53-Dependent Embryonic Phenotypes

In mouse embryonic stem cells and early embryos, *p53* restricts expression of the DNA methyltransferases *Dnmt3a* and *Dnmt3b* while up-regulating *Tet1* and *Tet2*, which promote DNA demethylation. Hence, *p53* helps maintain DNA methylation homeostasis and clonal homogeneity, a function that may contribute to its tumor suppressor activity [40]. Despite the fact that *p53* is not essential for mammalian development, *p53*-/- embryos can display developmental phenotypes [41]. Depending on their genetic background, some *p53*-/- embryos exhibit exencephaly [42,43], defects in eyes and teeth [43,44,45], and reduced pregnancy rates and litter sizes that result from decreased uterine levels of LIF interleukin-6 [46]. These effects reflect the fact that, although the *p53* protein is largely inactive in stem cells, when activated, *p53* helps to commit these cells to developmental lineages through a series of epigenetic changes [7].

## 3. What Next?

Although much has been learned, questions still remain unresolved due largely to two facts: (1) *p53* affects expression of hundreds of different genes required to maintain cell viability. (2) *p53* is not essential for mammalian development. Therefore, the question is not whether *p53* is required for a specific event to occur during development, but whether or not *p53* contributes significantly to that event.

When does *p53* activity first appear during mammalian development? Does *p53* affect genomic stability in embryonic cells as it does in adult cells? Does *p53* help maintain a pluripotent state? Does *p53* facilitate apoptosis or cell differentiation?

## 4. *p53* Activity Begins in Pluripotent Stem Cells and Becomes Critical with Gastrulation

Presumably, *p53* maintenance of genomic stability and induction of the WNT signal transduction pathway require *p53*-dependent transcription of specific genes. Thus, the question arises as to when *p53*-dependent transcription become significant during embryo development. 

### 4.1. p53 Activity Is First Detected at the Late Blastocyst Stage

Transgenic mice in which ectopic expression of a EGFP reporter gene is driven by a *p53*-dependent response element from either the cyclin-dependent kinase inhibitor *Cdkn1a*/*p21* or the pro-apoptotic *p53* upregulated modulator of apoptosis *Bbc3/Puma* promoter demonstrate that *p53* transcriptional activity exists as early as late stage blastocysts and early post-implantation epiblast [47] (Figure 3). Furthermore, double-strand DNA breaks introduced by X-irradiation of embryos at either E3.5 (blastocysts) or E9.5 (organogenesis) revealed that *p53*+/+ embryos die more frequently than *p53*-/- embryos, whereas *p53*-/- embryos exhibit more developmental anomalies [48]. X-irradiated *p53*+/+ embryos undergoing organogenesis contain a greater number of apoptotic cells than *p53*-/- embryos. *p53* appears to facilitate apoptosis in response to double-strand DNA breaks induced by X-irradiation of mouse embryos, but only after they developed to late stage blastocysts (E5) [49]. No significant change in cell proliferation was observed following X-irradiation, but wild-type blastocysts exhibited 2 to 3-times more apoptotic cells than *p53*-/- blastocysts. Apoptosis, defined as TUNEL-positive DNA-positive cell fragments, occurred primarily in the inner cell mass. Thus, *p53* transcriptional activity and DNA damage is evident as early as the late blastocyst stage and increases during organogenesis.

Nevertheless, *p53*-dependent transcription of endogenous genes in ESCs is suppressed by multiple mechanisms. The bulk of *p53* protein is localized to the cytoplasm in naïve ESCs, and *p53* activity is maintained at low levels by MDM2 [50,51,52]. SIRT1 prevents translocation of cytoplasmic *p53* to the nucleus [53], and AURKA inactivates *p53* by phosphorylating it [54]. 

This apparent contradiction between ectopic EGFP expression and endogenous gene expression reflects the fact that the number of transcripts produced from the EGFP reporter gene, and the number of transcripts produced from the endogenous *p21* and Puma genes were not determined. Thus, the simplest explanation is that endogenous *p53*-dependent transcription at the beginning of mouse development is low in pluripotent embryonic cells compared with differentiated embryonic cells, in which case the primary role of *p53* is to facilitate the transition from a pluripotent state to a differentiated state. In support of this hypothesis, ESC differentiation responds to changes in the cellular level of *p53*. In wild-type ESCs, the MDM2 specific inhibitor nutlin induces ESC differentiation as a consequence of *p53* activation, *p21* up-regulation and cell cycle arrest [55,56].

### 4.2. p53 Activity Is Regulated Post-Translationally and Cannot Induce Lethality until Gastrulation

Under normal conditions, *p53* expression is very low; it is a short-lived protein whose stability and activity are regulated by post-translational modifications such as phosphorylation, methylation and acetylation that are critical for *p53* activation, and by association with specific *p53* regulatory proteins such as MDM2 and MDM4/MDMX [57,58,59]. *MDM2* is a *p53* target gene that encodes a ubiquitin ligase that mediates degradation of *p53*, thereby creating a feed-back loop that regulates *p53* activity. MDM4 Inhibits *p53* by binding to its transcriptional activation domain. MDM4 can reverse MDM2-targeted degradation of *p53* while maintaining suppression of *p53* transactivation and apoptotic functions. In addition, the ‘retinoblastoma-binding protein’ RBBP6/PACT [60,61,62] and the ‘protein activator of interferon induced protein kinase EIF2AK2’ PRKRA/RAX/PACT [63,64] have been reported to regulate *p53* activity. 

Gastrulation occurs from E6.25 to E9.5 wherein the epiblast gives rise to the embryo proper through differentiation into the three primary germ layers, ectoderm, mesoderm and endoderm [65]. *p53* activity is suppressed in mice until post-implantation development by virtue of the interaction between *p53* and MDM2, MDM4 and PACT/RBBP6 [57,61]. Inactivation of the *MDM2*, *MDM4* or *PACT/RBBP6* gene in mouse embryos results in lethality, but only in the presence of *p53* protein. For *Mdm2-/-* embryos, demise occurred after implantation of the embryo in the wall of the uterus but before day 7.5 of gestation (taken as E5.5) [66,67]. Furthermore, the presence or absence of normal physiologic levels of Mdm2 in embryonic fibroblasts isolated from 12 to 14 day old embryos has no effect on the growth or tumorigenic potential of *p53*-deficient mouse embryonic fibroblasts, indicating that *Mdm2* does not have a *p53*-independent role in regulating cell proliferation [68,69]. For *PACT/RBBP6-/-* embryos, lethality after implantation but before E7.5 [61]. For *MDM4*, lethality occurs between E7.5 and E12 [70,71,72]. All three phenotypes were rescued by transferring the mutated *p53* negative regulator gene to a *p53*-nullizygous mouse background, in which case mice develop normally. Thus, embryonic death in the absence of *MDM2*, *MDM4* or *PACT/RBBP6* results from activation of *p53* protein. 

### 4.3. Take-Home Lesson

*p53* activity is first detectable during the late blastocyst stage and confined primarily to pluripotent cells. However, *p53* expression is too low to induce either cell cycle arrest or cell death upon release from post-translational regulation by MDM2 or PACT/RBBP6 until after the blastocyst has implanted (E4.5) and gastrulation has begun (E6.25-E7.5, Figure 2). However, DNA damage begins to accumulate in *Mdm2-/-* blastocysts, but less so in embryos that are null for both *Mdm2* and the *p53*-dependent pro-apoptotic gene *Bax*, suggesting that unregulated *p53* can initiate apoptosis in blastocysts that is not lethal until gastrulation [73]. Nevertheless, in the absence of both MDM2 and BAX proteins, embryonic lethality still occurs during E6.5-E7.5, due to arrest of cell proliferation (presumably senescence). In contrast, *Mdm4-/-* embryo lethality results simply from *p53*-dependent arrest of cell proliferation (presumably senescence) [70,71,72]. 

## 5. *p53* Maintains Genomic Stability during Embryonic Development

The earliest demonstration of *p53*-dependent maintenance of genomic stability occurs during embryonic stem cell proliferation when *p53* prevents accumulation of polyploid and aneuploid cells [74]. Aneuploidy is a hallmark of cancer [10]. In the absence of *p53*, polyploid and aneuploid cells arise during the 21 days required for gestation in mice. However, only a fraction of these cells will produce a cancer, thereby accounting for the fact that *p53*-/- mice do not die until they succumb to tumors as adults.

### 5.1. Tetraploidy -> Polyploidy -> Aneuploidy

Tetraploidy is a precursor to polyploidy and polyploidy leads to aneuploidy [75]. Tetraploidy is an aberrant event that results from endoreplication (multiple S-phases without an intervening mitosis), cytokinesis failure, or mitotic slippage. When that occurs, wild-type mouse embryonic fibroblasts or human fibroblasts are arrested in G1 phase with 4N DNA content, whereas *p53*-/- cells re-enter the cell cycle and initiate another round of DNA replication [76,77]. *p53*-dependent cell cycle arrest requires the cyclin-dependent kinase inhibitor *p21*. The same effects are also observed with human fibroblasts, except that tumor suppressor ‘retinoblastoma transcriptional corepressor 1’ (RB1) also plays a role. Thus, the *p53* dependent checkpoint following disruption of the mitotic spindle functionally overlaps with the *p53*-dependent checkpoint following DNA damage. In addition, *p53* loss dysregulates the spindle assembly checkpoint by up-regulating *MAD2*, which increases chromosome missegregation and tetraploidization [78]. In the context of tetraploid cells, *p53* loss leads to an increased rate of multipolar mitoses and subsequent chromosome missegregation [79]. Consequently, loss of *p53* drastically accelerates tumor development in aneuploid cells [80].

### 5.2. Teratoma Formation

Teratomas are solid tumors composed of differentiated cells derived from all three primary germ layers [81]. Both *p53*+/+ and *p53*-/- mouse ESCs produce teratomas at ectopic sites [82,83]. During self-renewal, *p53*-/- ESCs develop a high incidence of karyotype abnormalities [82], and although they retain their ability to contribute to normal development in chimeric embryos, *p53*-/- ESCs exhibit defects during differentiation [84]. Thus, *p53* is not required for ESC differentiation, because both *p53*+/+ and *p53*-/- mouse or human ESCs can differentiate in vitro [82,85]. However, *p53* does act as a ‘guardian of the genome’ by inhibiting ESCs with abnormal karyotypes from proliferating. Fibroblasts isolated from *p53*-/- mouse embryos and then cultured in vitro readily accumulate aneuploid cells [86]. The absence of *p53* is not sufficient to cause aneuploidy in vivo, although cells without *p53* are prone to accumulate abnormal chromosomes after oncogene activation [87]. 

### 5.3. Take-Home Lesson

In the absence of detectable *p53* transcriptional activity prior to the appearance of embryonic stem cells, *p53* begins its role as ‘guardian of the genome’ during the late blastocyst stage and continues to the adult.

## 6. Maintaining the Pluripotent State

The role of *p53* in maintaining a pluripotent state was first suggested by the fact that *p53* gene expression is greater in cells derived from 10–14 days old mouse embryos than in cells derived from 16 days old mouse embryos [88]. Subsequent studies confirmed that expression of *p53* was high in naïve ESCs [50], downregulated during embryogenesis [88,89], and virtually absent in terminally differentiated cells [90]. More recently, pifithrin-α (PFT-α), an inhibitor of *p53*-dependent transcription, inhibited cell number, colony size, cell cycle progression and DNA synthesis in ESCs, suggesting that in the absence of stress, *p53* is required for ESC self-renewal [91]. However, PFT-α inhibition of transcription is highly dependent on the *p53* target gene selected, and PFT-α is not an effective antagonist of *p53* activation by nutlin inhibition of Mdm2 [92]. 

The pluripotent state is currently attributed to expression of 10 essential transcription factors in human and mouse ESCs [93,94,95,96,97]. They are the ‘core genes’ *Pou5f1p5/Oct4*, *Sox2*, and *Nanog*, along with *Tcf3/E2a*, *Klf4*, *Myc/c-Myc*, *Esrrb*, *Sall4*, *Tbx3*, and *Stat3*. Remarkably, only *Nanog* and *Myc* are among the 343 genes considered to be regulated directly by the *p53* transcription factor [32], and both are down-regulated by *p53* [23,24,98]. Therefore, if *p53* facilitates maintenance of the pluripotent state, it does so indirectly. 

### 6.1. Pluripotent Stem Cells Maintain p53 at Low Levels

In both ESCs and induced pluripotent stem cells (iPSCs), *p53* activity is maintained at low levels through post-translational modifications by ubiquitylation, acetylation, phosphorylation, methylation or sumoylation of specific residues in *p53* protein [99]. For example, ubiquitylation by MDM2 and TRIM24 maintain low levels of *p53* in human ESCs [100]. Phosphorylation of *p53* by AURKA at S212 and S312 inhibits differentiation of ESCs [54]. Deacetylation of *p53* at K373 by the NAD-dependent deacetylase SIRT1 inhibits nuclear *p53* activity [101]. 

### 6.2. Pluripotency Is Maintained through p53 Regulation of LncRNA Expression

Long non-coding RNAs (LncRNAs), >200 nucleotides with poly-adenylated tail but without an open reading frame, are highly expressed in human pluripotent ESCs [102,103]. Repression of LncRNAs promotes ESC differentiation [102]. More than 40 LncRNAs are expressed in human ESCs, and they are regulated by *p53* to maintain a balance between pluripotency and differentiation. 

For example, the LncRNAs termed lncPRESS1 and lncPRESS2 maintain pluripotency whereas the LncRNA termed HOTAIRM1 promotes differentiation [104]. lncPRESS1 prevents chromatin localization of SIRT6, which results in histone acetylation at H3K56 and H3K9 at the promoters of core pluripotency genes, thereby maintaining their transcription [104]. In human ESCs, H3K56 acetylation is associated the transcriptional activation of core pluripotency genes [105]. lncPRESS4, also known as TUNA (Tcl1 upstream neuron-associated or linc86023), is required for maintenance of pluripotency by directly binding to the promoters of the *Nanog*, *Sox2* and *Fgf4* genes [106].

### 6.3. Pluripotency Is Maintained through WNT Signaling

Under stress conditions, *p53* stimulates ESCs to secrete WNT ligands that inhibit differentiation, thereby maintaining a pluripotent state [107]. As *p53* activates the expression of *LIF interleukin-6* [108], the WNT signaling pathway and its synergistic interaction with LIF interleukin-6 play important role in maintaining pluripotency and self-renewal of human and murine cells [109,110]. 

### 6.4. Micro-RNAs Maintain Pluripotency by Inactivating p53

MicroRNAs (miRs) repress the expression of mRNA targets by promoting translational repression and mRNA degradation. [111]. Their regulatory effects appear complex. miR-294 and miR-302 promote the abbreviated G1 phase of the ESC cell division cycle and suppress ESC differentiation [[112] and references therein]. Other miRNAs can prevent ESC self-renewal, and their activities are antagonized by miR-294 and miR-302. The differentiation-inducing effect of miRNAs was retinoblastoma protein (Rb) dependent, but their ability to inhibit ESC self-renewal was Rb-independent. 

In contrast to wild-type ESCs, elevated levels of *p53* activity restricts differentiation of ESCs that are deficient in miRs [113]. *p53* prevented these ESCs from differentiating in the neural direction. However, expression of miR-302 promoted ESC differentiation by direct interaction with *p53*. Similarly, inactivation of *p53* by association with SV40 large T antigen, by suppressing *p53* RNA translation with shRNA, or by genetic ablation of the *p53* gene enabled differentiation. In contrast, activating *p53* by treating miR-deficient ESCs with the MDM2 inhibitor nutlin inhibited neural differentiation. Thus, low levels of *p53* allowed differentiation, whereas high levels of *p53* inhibited differentiation. Moreover, cellular levels of *p53* appear to be regulated by various miRs, presumably in response to environmental signals.

### 6.5. Pluripotency Is Maintained through p53 Isoforms

A transactivation-deficient isoform of *p53*, Δ40*p53* is highly expressed in ESCs, and is the major *p53* isoform during early mouse embryonic development. *p53* is maintained in an inactive state by the presence of Δ40*p53*. Haploinsufficiency for Δ40*p53* causes a loss of pluripotency in ESCs and compromises their ability to grow, while its increased dosage prolongs pluripotency and inhibits differentiation. Δ40*p53* controls *p53* at targets such as *Nanog* and *the IGF-1* receptor to switch from pluripotent to differentiated cells [114]. Δ40*p53* overexpression reduces *p53* activity by inhibiting its transactivation domain [115,116] and promoting nuclear export of *p53*–Δ40*p53* heterotetramers [117]. In ESCs, overexpression of Δ40*p53* interferes with the ability of *p53* to regulate the transcription of *p21*, *Nanog*, and *Igf-1R* which are involved in the switch between pluripotency and differentiation [23,118,119]. Moreover, N-terminal truncated isoforms Δ*Np73* of *p73* decreases *p53* leading to an increase in *Nanog* gene expression which subsequently enhanced human iPSC generation [120]. Although less explored in stem cell regulation, *p63* and *p73* functions in cooperation with *p53* in the regulation of adult stem-like cells [121].

### 6.6. Take-Home Lesson

*p53* does not regulate expression of genes essential for ESC self-renewal. In fact, *p53* activity must be maintained at a low level in ESCs to prevent differentiation (Section 7). Moreover, *p53* activity is not a significant factor in maintaining the pluripotent state. Both wild-type and *p53*-/- ESCs undergo self-renewal in vitro as well as *in vivo*. They both proliferate, produce teratomas, contribute to chimeras, and produce living mice. In fact, *p53*-/- ESCs proliferate faster [84] and *p53*-/- 2-cell embryos development faster [49] than their wild-type counterparts. The primary distinction between wild-type and *p53*-/- mice is that *p53*-/- mice accumulate genomic abnormalities during development that eventually result in tumors and lethality. 

## 7. Pluripotent Stem Cell Differentiation

### 7.1. p53 Facilitates ESC Differentiation

*p53* is not essential for ESC differentiation, because embryos lacking *p53* genes can still gastrulate and develop, and newborns can thrive until succumbing to tumors at 3 to 5 months of age [14]. Nevertheless, *p53* facilitates differentiation of pluripotent stem cells [40,84,104], and *p53* inhibits conversion of differentiated cells into pluripotent cells [122,123,124,125]. 

The frequency at which reprogramming of differentiated cells into pluripotent cells by ectopic expression of genes, such as *Oct4*, *Sox2*, *Nanog* and others is suppressed by the high levels of *p53* in differentiated cells and stimulated by reducing *p53* expression, because *p53* represses transcription of the reprogramming genes [23,24]. However, differentiation of ESCs is not induced simply by *p53* inhibiting expression of the core pluripotency genes *Oct4*, *Sox2*, or *Nanog*, because these genes are down regulated during LIF-deprivation at equivalent rates in both *p53*+/+ and *p53*-/- ESCs [126]. Therefore, if *p53* facilitates differentiation of ESCs, it must do so indirectly as well as directly. 

### 7.2. DNA Damage Induces p53-Dependent Differentiation

Doxorubicin/Adriamycin increases the number of differentiated colonies appearing in naïve *p53*+/+ ESCs, but not in *p53*-/- ESCs, due to *p53*-mediated transcriptional repression of ESC-specific transcription factors [24]. Doxorubicin also dramatically stimulated expression of genes characteristic of differentiated cells in *p53*+/+ embryoid bodies, but not in *p53*-/- embryoid bodies. ESC aggregation is the initial step in triggering ESC differentiation in the form of ‘embryoid bodies’ that consist of cells characteristic of cells derived from the three primary germ layers [127,128]. Thus, the ability of DNA damage to promote ESC differentiation is *p53*-dependent. 

### 7.3. p53 Can Inhibit Pluripotency

In response to DNA damage, *p53* protein is activated by phosphorylation at S315 and over expressed. Activated *p53* binds to the *Nanog* promoter where it displaces OCT4 and SOX2, thereby suppressing *NANOG* expression and driving ESCs differentiation [23,129,130]. *p53* also mediates direct repression of *Nanog* transcription by recruitment of MSIN3A to *p53* bound at the *Nanog* promoter [23]. MSIN3A is a histone deacetylase that represses transcription [131]. Activation of *p53* by treating cells with nutlin, a specific inhibitor of MDM2, also induces ESC differentiation [55]. Nutlin induced suppression of *NANOG* and *OCT4* expression with concomitant induction of *GATA4* and *GATA6* expression, two transcription factors that are essential for embryonic development during gastrulation [132]. 

### 7.4. p53 as A Barrier to Reprogramming

Reprogramming differentiated cells into ‘induced pluripotent stem cells’ is achieved specifically by over expressing the core pluripotency transcription factors OCT4, SOX2 and NANOG, in combination with other transcription factors, such as KLF4, LIN28A/LIN28 and MYC [133]. *p53* acts as barrier to reprogramming by inhibiting expression of these genes in differentiated cells. This is evident from the fact that reprogramming *p53*-/- somatic cells is much easier than reprogramming *p53*+/+ somatic cells [122,123,124,125]. Apart from this, MYC activates ARF-dependent and *p53*-dependent pathways that elevate the *p53* barrier to reprogramming [134]. Activation of *p53*-dependent pathways in response to DNA damage also elevate expression of *p53*, which suppresses expression of *NANOG* [135].

### 7.5. p53-Dependent Formation of Haploid Cells

Haploid cells are a characteristic of early post-implantation embryos [136]. Deprivation of LIF interleukin-6 forces naive ESCs to either differentiate in the mesendodermal direction or undergo apoptosis [137,138]. Remarkably, LIF-deprivation of naïve ESCs drives formation of haploid-like cells in *p53*+/+ cells, but not in *p53*-/- cells [126]. Haploid ESCs are a useful tool for loss-of-function genetic screening [139].

### 7.6. Take Home Lesson

*p53* promotes ESC differentiation, and once a differentiated state is produced, *p53* activity stabilizes the differentiated state.

## 8. DNA Damage Response in ESCs

### 8.1. Physiological States of Pluripotent Embryonic Stem Cells

ESCs exhibit three physiological states termed ‘ground state’, ‘naïve’, and ‘primed’ [140,141,142]. ESCs cultured in the presence of serum and LIF interleukin-6 are considered ‘naïve ’ pluripotent ESCs, because they give rise primarily to somatic tissues and germ cells but not to the trophectoderm. Most studies on ESCs fall into this category (Figure 2). 

‘Naïve ESCs’ cultured in defined medium (no serum) containing two kinase inhibitors, one against MAP2K1/MEK1 and one against FRAT2/GSK-3, are thought to model ‘ground-state’ pluripotent ESCs, because they can give rise to both extraembryonic (placental and yolk sac) and embryonic (epiblast and its derivatives) cells. ‘Ground-state ESCs’ are termed 2iESCs [143]. 2iESCs are pluripotent and perform well in chimera formation [144].

‘Naïve ESCs’ can also differentiate into ‘primed ESCs’ when cultured in the presence of activin and fibroblast growth factor. Both ‘naïve ESCs’ and ‘primed ESCs’ can differentiate into mesoderm, endoderm, ectoderm, as well as germ cells. However, ‘naïve ESCs’ can readily generate chimeric animals, whereas ‘primed ESCs’ cannot. Therefore, primed ESCs are thought to model the epiblast in early embryos and are termed EpiSCs. In fact, transcriptomes in mouse EpiSCs derived from the pre-gastrula stage to late-bud-stage embryos are hierarchically distinct from those of ESCs and epiblast ectoderm but similar to the ectoderm of the late-gastrula embryo [145]. Thus, naïve ESCs are thought to model the pre- or peri-implantation embryonic epiblast, whereas EpiSCs model the early post-implantation epiblast [146,147].

### 8.2. Cell Cycle Arrest in Naïve ESCs Is Not p53-Dependent

The absence of a G1-checkpoint together with the presence of a G2-checkpoint is a hallmark of both naïve ESCs and cancer cells [148]. The G2 checkpoint appears as a transient accumulation of cells containing 4N DNA in response to DNA damage prior to induction of apoptosis [149]. Double-strand DNA breaks induced by culturing cells with the topoisomerase II inhibitor doxorubicin/Adriamycin activates the G2 checkpoint in naïve ESCs regardless of the presence or absence of *p53* [24,126,150,151], *p21* or PUMA [[126], Figure 4C)].

The G1 checkpoint is a more nuanced response to cell stress that retards entrance into S phase [152]. Naïve mouse ESCs lack a G1 DNA damage checkpoint [126,150,151,153,154,155], although it might exist in 2iESCs [156]. Naive ESCs are characterized by hyper-phosphorylated RB1 protein, lack of G1 control, and rapid progression through the cell cycle. In contrast, 2iESCs have a longer G1-phase with hypophosphorylated RB1, implying that they have a functional G1 checkpoint. The RB1-dependent G1 restriction point is active in ESCs propagated under 2i culture conditions but abrogated by ERK-dependent phosphorylation when cultured in serum [157].

### 8.3. Apoptosis in Naïve ESCs Is Not p53-Dependent

Given that *p53* activation in mice does not induce cell death until gastrulation, and that *p53* is not required to activate the G2 checkpoint, one would expect that pluripotent stem cells would not require *p53* to induce apoptosis in response to DNA damage or other cell stresses. Remarkably, of the nine studies that have addressed this issue, three concluded that *p53* is not required for DNA damage to induce cell death [126,151,153] and six concluded that *p53* is required [150,155,158,159,160]. Six studies compared *p53*+/+ with *p53*-/- ESCs [126,151,153,155,158,159]. Two studies used shRNA to constitutively suppress *p53* expression [150,160]. One study relied solely on changes in *p53* expression in response to apoptotic stimuli [52].

Since pluripotent stem cells are programmed to either proliferate, differentiate, or die, Jaiswal and coworkers [126] investigated multiple parameters that might affect naïve ESCs. They found that, regardless of their derivation, naïve ESCs do not require *p53*, *p21* or *PUMA* either to activate the G2-checkpoint or to undergo programmed cell death rapidly and efficiently via a non-canonical apoptosis pathway. The effects of the concentration of the DNA damaging agent commonly used in these studies (doxorubicin/Adriamycin) and cell confluency were marginal, but the effects of cell differentiation were dramatic; *p53*-dependent regulation of cell division and apoptosis were acquired during *p53*-dependent differentiation of ESCs in vitro.

To eliminate the possibility that these conclusions depended on either the source or derivation of ESCs, wild-type and *p53*-/- ESCs derived directly from blastocysts were characterized in parallel with ESCs in which the *p53* genes were ablated in vitro. The effects of doxorubicin in these two ESC derivations were indistinguishable over a 40-fold range of concentrations, and cell culture conditions were compared over a 50-fold range of seeding densities. To eliminate methodology-dependent biases, apoptosis was quantified by time dependent increases in annexin-V binding to detect apoptosis, in propidium iodide staining to distinguish apoptosis from necrosis, and in trypan blue staining to distinguish live cells from dead cells, as well as time dependent loss of DNA to establish cell death. Analyses of *p53*, *p21*, PUMA, γ-H2AX, PARP, and CASP3 proteins were done to confirm genotypes, DNA damage, cell cycle arrest and apoptosis. Canonical apoptosis involves cleavage of CASP3, whereas non-canonical apoptosis does not. Non-canonical apoptosis was confirmed by translocation of AIFM from the cytoplasm to the nucleus.

The inhibitory effect of only 50 nM doxorubicin on naïve ESCs-regardless of the presence or absence of *p53*—is evident from visual inspection of cultured cells (Figure 4A,B). The lethal effect of doxorubicin is evident from the accumulation of cells with <2N DNA content (Figure 4C,D). Short exposure (24 h) of low concentration (50 nM) of doxorubicin to mouse ESCs, then allowing them to recover for 96 h proved that even minimal DNA damage is enough to induce apoptosis in mouse ESCs regardless presence of *p53*.

### 8.4. DNA Damage Response in Ground-State 2iESCs

Studies on the role of *p53* in ground-state 2iESCs produced contradictory results that mirror those reported for naïve ESCs. Two studies in mouse 2iESCs revealed that DNA damage induced either by aphidicolin inhibition of DNA polymerase-α or doxorubicin inhibition of topoisomerase II activates genes characteristic of zygotic gene expression in mouse 2-cell to 4-cell embryos [161,162]. Both studies concluded that the mechanism driving this expression is mediated by an *ATR* and *CHK1* response to double-strand DNA breaks. However, one study concluded that activation required *p53* expression [162], whereas the other study concluded that it did not [161]. Critical experiments in which *p53*+/+ and *p53*-/- 2iESCs were compared were carried in both studies, but with opposite results. A third study concluded that doxorubicin induced *p53*-dependent apoptosis in 2iESCs [156].

### 8.5. Reconciling Disparate Data

#### 8.5.1. Bases for Reconciliation

Contradictory results concerning the DNA damage response during early mammalian development can be reconciled on the following bases: (1) Suboptimal culture conditions for ESCs can result in spontaneous differentiation or cell death. (2) Treatment of cells with compounds at far greater concentrations that required to affect a specific target invariably affects unintended targets that can produce unexpected side effects. (3) Analysis of nullizygous cell lines is imperative for determining whether or not a gene is required for a particular function. (4) Time dependent comparisons of *p53*-/- with *p53*+/+ cell lines are imperative to determine whether or not a gene is required to complete a particular function. (5) Constitutive suppression of genes with shRNA selects for clones that remain viable, thus selecting for off-target mutations in addition to the targeted gene.

#### 8.5.2. Reconciliation of Naïve ESC Data

Six of the studies described above compared ESCs derived from *p53*-/- blastocysts produced by mating *p53*+/- mice with *p53*+/+ ESCs from the same matings [61,126,151,153,155,159]. Remarkably, two studies using blastocyst derived-ESCs (BD-ESCs) from the same source (Rudolf Jaenisch, MIT, Cambridge, MA, USA) reported contradictory results. One found that 1.84 µM Doxorubicin (ADR) rapidly induced apoptosis in >90% of *p53*+/+ and *p53*-/- cells [153], whereas the other reported that the same concentration of ADR induced apoptosis in 50% of *p53*+/+ cells but 0% of *p53*-/- cells [159].

The simplest explanation is that culturing ESCs to “sub-confluence” before adding ADR [159] created conditions in which excessively high concentrations of drug induced apoptosis in *p53*+/+ cells more rapidly than in *p53*-/- cells, as shown in other studies [126,153]. For example, ESCs undergoing self-renewal should not be allowed to proliferate to >80% confluency, because clumps of *p53*+/+ ESCs tend to differentiate [163], and *p53*-dependent events are clearly associated with differentiated cells.

Depending on seeding density and ADR concentration, ADR triggers *p53*-dependent apoptosis an average of 8.4 ± 0.5 h earlier than *p53*-independent apoptosis, but both *p53*+/+ and *p53*-/- naïve ESCs complete apoptosis within 72 h (Figure 4D). As little as 0.05 µM Doxorubicin is sufficient to induce apoptosis at the same time in both *p53*+/+ and *p53*-/- naïve ESCs which then proceeds to completion at equivalent rates. The fact that ectopic over-expression of certain *p53* mutations also suppressed ADR-induced apoptosis [159] simply reflects the fact that *p53* affects expression of hundreds of different genes, some of which affect apoptosis. Many naturally occurring *p53* mutations have the opposite effect; they gain additional oncogenic functions that endow cells with growth and survival advantages [164]. In addition, culture conditions are critical to maintain the pluripotent ESC state [141] and to prevent DNA damage from rapidly accumulating in ESCs cultured under suboptimal conditions [165]. Suboptimal culture conditions might well contribute to conflicting results.

Remarkably, a second contradiction emerged from two studies using same source of ESCs (Yang Xu, Univ. California, San Diego, CA, USA). Both studies concluded that *p53* is not required for cell cycle arrest [126,155] and their results with *p53*+/+ ESCs are indistinguishable, regardless of cell seeding density or ADR concentration [126,155]. However, one study concluded that *p53* is essential for ADR-induced apoptosis [155] whereas the other study concluded that efficient apoptosis induced either by ADR or other stress inducers is not *p53*-dependent [126]. Reconciliation is achieved by virtue of the fact that results with blastocyst derived ESCs (BD-ESCs) were virtually indistinguishable from results with *p53*-/- ESCs derived by ablating the *p53* genes in ESCs homozygous for a conditional *p53* gene knockout [126]. Moreover, Li and co-workers relied on CASP3 cleavage to confirm apoptosis, which they detected with a monoclonal antibody specific for the cleaved form. Thus, they did not recognize that the extent of CASP3 cleavage was insignificant. Moreover, the time delay for initiation of apoptosis exhibited by naïve ESCs cultured with high concentrations of ADR meant that cleaved CASP3 did not appear until after 24 h. CASP3-dependent apoptosis also rapidly declines in *p53*-/- ESCs cultured under stress, such as the extremely high seeding density (260,000 cells/cm^2^) used in [155]. This was not a problem using the seeding densities in [126].

Remarkably, two reports from the same laboratory reported apparently conflicting results. One report on induced UV-dependent DNA-damage in *p53*-deficient ESCs indicated that apoptosis is *p53*-independent [151], whereas another report from the same laboratory concluded that apoptosis is *p53*-dependent [158]. This conundrum likely resulted from the fact that the *p53*-deficient cells in these studies exhibited a complex karyotype that might include additional genetic defects. Neither report provided evidence that the cells used were pluripotent stem cells.

Another study did not compare *p53*+/+ cells with *p53*-/- cells [52]. Instead, their conclusion that *p53* is required for apoptosis in naïve ESCs was based solely on changes in *p53* expression in response to apoptotic stimuli. Such changes in *p53* expression are equally consistent with a role for *p53* in cell differentiation. ESCs under stress characteristically undergo either differentiation or apoptosis [55,104], and changes observed in gene expression and relocalization of *p53* from cytoplasm to nucleus are characteristics of ESC differentiation as well as apoptosis.

Two studies concluded that ADR or etoposide-induced apoptosis was *p53*-dependent in ESC clones where either human *TP53* [160] or mouse *Tr**p53* [150] was suppressed constitutively by shRNA. Reconciliation comes from three possibilities. First, neither study excluded the possibility that off-target effects also repressed expression of genes required for apoptosis. Second, selection of viable ESC lines in these studies would select clones resistance to apoptosis, because constitutive suppression of *p53*-expression by shRNA in mouse ESCs and embryos promotes clonal heterogeneity by disrupting DNA methylation homeostasis [40]. Third, since ESCs under stress characteristically undergo either differentiation or apoptosis [55,104], changes observed in gene expression and relocalization of *p53* from cytoplasm to nucleus are characteristics ESC differentiation as well as apoptosis.

#### 8.5.3. Reconciliation of Ground-State 2iESC Data

Three studies have reported the effects of doxorubicin on mouse *p53*+/+ and *p53*-/- 2iESCs. One study cultured 2iESCs with 1 µM doxorubicin for 6 h and concluded that the DNA damage response was *p53*-dependent [162]. Another study carried out the same experiment using 1 µg/mL (1.84 µM) doxorubicin for 48 h and concluded that the DNA damage response was *p53*-independent [161]. Still a third study cultured 2iESCs with 1 μM doxorubicin for 16 h and observed that *p53*+/+ cells underwent apoptosis more quickly than *p53*-/- cells (63% *p53*+/+ cells versus 13% *p53*-/- cells) [156]. These results could easily be reconciled if the experiments did not use excessively high concentrations of doxorubicin and if they monitored the effects of doxorubicin over time.

Similar conditions with naïve ESCs revealed that apoptosis begins more quickly in *p53*+/+ cells than in *p53*-/- cells, particularly with excessively high concentrations of doxorubicin, but the rates at which cells undergo apoptosis are equivalent, and the extent of apoptosis within 48 to 72 h is equivalent [126,153]. As little as 0.05 µM doxorubicin is sufficient to induce apoptosis in naïve ESCs (Figure 4). Depending on seeding density and doxorubicin concentration, *p53*+/+ ESCs initiate apoptosis 8.4 ± 0.5 h earlier than *p53*-/- ESCs, but both *p53*+/+ and *p53*-/- ESCs complete apoptosis within 72 h (Figure 4D). To determine whether or not *p53* is essential for a DNA damage response in ground-state 2iESCs requires that the response of 2iESCs to different extents of DNA damage is characterized over time.

A separate concern is that 2iESC culture conditions enforce self-renewal and a dramatic loss of spontaneously differentiating cells (Navarro, 2018). So far, only naïve ESCs can proliferate and transit easily to the ‘ground-state’ under these conditions, whereas neither EpiSCs nor differentiated somatic cells survive these conditions. In addition, the application of CRISPR-Cas9 technology to isolate ESC clones with specific gene alterations raises the possibility that additional ‘off-target’ mutations might be included that affect gene expression, cell proliferation, or cell death.

### 8.6. Take-Home Lesson

Cell cycle arrest in naïve ESCs is *p53*-independent. Similarly, both the rate and the extent of apoptosis in naïve ESCs is *p53*-independent. However–depending on experimental conditions–*p53* can accelerate initiation of apoptosis by a few hours. Recognizing this distinction requires both time dependent assays and DNA damaging agent concentration dependent assays. Results with naïve ESCs appear to be true for 2iESCs as well, and is likely to be true for primed EpiSCs, although this hypothesis has not yet been tested. However, these conclusions are consistent with the fact that inactivation of any one of the three proteins essential for post-translational regulation of *p53* activity (Section 1), does not arrest embryonic development until gastrulation or later (Section 4.2). Therefore, the level of *p53* activity in vivo is not sufficient to induce cell cycle arrest or apoptosis until gastrulation.

## 9. DNA Damage Response in Differentiated Cells

### 9.1. A p53-Dependent DNA Damage Response Is Acquired during ESC Differentiation

Depriving naïve ESCs of LIF interleukin-6 forces them to either differentiate in the mesendodermal direction or undergo apoptosis [137,138]. Under these conditions, LIF-deprivation of *p53*+/+ ESCs increased genomic stability, suppressed the DNA damage induced G2-checkpoint, and increased the rate and extent of DNA damage-induced apoptosis [126]. The G2-checkpoint in ESCs was *p53*-independent, but after ESC differentiation in vitro, the G2-checkpoint was evident only in *p53*-/- cells; differentiation of *p53*+/+ ESCs resulted in loss of the G2-checkpoint. Moreover, apoptosis in either *p53*+/+ or *p53*-/- ESCs included cleavage of caspase-3, whereas apoptosis in differentiated ESCs did not. Thus, LIF-deprivation of naïve ESCs in vitro transformed pluripotent cells into differentiated cells with characteristics similar to oncogenic MEFs.

### 9.2. Viability of Embryonic Fibroblasts Is p53-Independent

Early passage *p53*-/- mouse embryonic fibroblasts (MEFs) isolated from E12 to E14 divide faster than wild-type MEFs, achieve higher cell densities, and have a higher fraction of division-competent cells at low cell density. The fraction of *p53*-/- MEFs with 2N DNA content (G1 phase) is less than in wild-type MEFS, consistent with a *p53*-dependent G1 checkpoint. *p53*+/+ MEFs senesce in response to excessive DNA damage, whereas *p53*-/- MEFs develop aneuploidy and chromosomal abnormalities Thus, loss of *p53* in MEFs confers a proliferation advantage, but not immortality [86].

In contrast with embryos undergoing gastrulation, deletion of *Mdm2* has no effect on viability of MEFs [68,69]. Moreover, *p53*-/- MEFs and *p53*-/-; *Mdm2*-/- MEFs were indistinguishable in their proliferation, response to genotoxic agents, genetic instability, and ability to form spontaneous tumors. Thus, in the absence of *p53*, deletion of *Mdm2* has no additional effects on cell proliferation, cell cycle control, or tumorigenesis in differentiated embryonic cells.

### 9.3. DNA Damage Induces p53-Dependent Senescence in Embryonic Fibroblasts

*p53* induces cellular senescence when cell cycle arrest is not enough to repair DNA damage. In cells with oncogenic activation, telomere shortening, or oxidative stress, cellular senescence prevents proliferation permanently, while retaining the cell’s function [30,166]. However, release from non-senescence temporary cell cycle arrest can lead to proliferation of cells with oncogenic potential and development of tumors [21].

Only *p53*+/+ MEFs upregulate expression of *p53*, *p21*, and *PUMA* in response to double-strand DNA breaks introduced by doxorubicin [126]. Nevertheless, MEFs respond to low levels of either natural or induced DNA damage by undergoing *p53*-dependent senescence rather than apoptosis [68,86,126,167], whereas *p53*-/- MEFs undergo apoptosis [126,168,169]. Cell senescence is preceded by *p53*-dependent complete cell cycle arrest wherein cells do not accumulate in a specific cell cycle phase, but simply stop proliferating in place [170]. In the absence of *p53*, MEFs respond to DNA damage with a transient G2 checkpoint followed by *caspase-3* independent apoptosis [126,168,169].

At high doxorubicin concentrations, MEFs exhibit *p53*-independent apoptosis [126,171]. Similarly, in the absence of *MDM2*, artificially high levels of *p53* induce apoptosis [172]. Thus, the choice between senescence and apoptosis appears to depend on the cellular level of activated *p53* protein.

### 9.4. Apoptosis Is p53-Dependent in Embryonic Fibroblasts Immortalized with an Oncogene

MEFs exhibit *p53*-dependent apoptosis or premature senescence when they are immortalized with an oncogene such as *adenovirus E1a* [173,174,175,176], *Myc* [177] or oncogenic *Ras* [169]. In contrast, normal *p53*+/+ MEFs undergo complete cell cycle arrest in response to double-strand DNA breaks [126,178,179,180], whereas *p53*-/- MEFs continue to proliferate until they undergo apoptosis [126,171].

### 9.5. Take-Home Lesson

Only when embryonic development has produced differentiated fibroblasts, has *p53* activity been demonstrated to significantly increase the rate and extent at which DNA damage induces cell cycle arrest to allow repair of DNA damage, and cell senescence to prevent proliferation of cells with DNA damage. Whether or not DNA damage can also induce apoptosis in embryonic cells remains to be determined.

## 10. Summary

Analysis of the of effects of *p53* on mammalian development reveals several surprising conclusions. Although *p53* mRNA has been detected from zygote to adult (Section 1 and Section 2.1), *p53* protein has not. In fact, evidence for *p53*-dependent transcription first appears during mouse development in the inner cell mass of late blastocysts, and the *p53*-specific regulatory protein MDM2 first becomes essential for mouse development when gastrulation begins by preventing *p53* activity from arresting embryonic development (Section 4.1 and Section 4.2). Thus, *p53* activity appears to begin with formation of pluripotent stem cells and thereafter accumulates as cells undergo differentiation, suggesting that totipotent blastomeres undergo cell cleavage in the absence of *p53* activity.

*p53*-mediated cell cycle arrest allows non-malignant as well as malignant cells to repair damaged DNA. When DNA repair is complete, cells can reenter the normal cell cycle. In contrast, when cells have serious DNA damage, *p53* exerts its pro-apoptotic function to eliminate these cells, thereby preventing transfer of damaged DNA to daughter cells [181,182,183]. This function of *p53* is first detected with embryonic fibroblasts where *p53* significantly increases the rate and extent at which DNA damage induces cell cycle arrest to allow repair of DNA damage, and cell senescence to prevent proliferation of cells with DNA damage while maintaining their metabolic contribution. Ablation of *Mdm2*, *Mdm4 or Rbbp6* at the beginning of mouse development results in post-implantation lethality, but whether the failure of embryos to continue development results from high levels of *p53* activity inducing cell cycle arrest, apoptosis or cell senescence is not clear. What is clear, is that *p53* activity must be regulated during gastrulation and subsequent embryonic development. Nevertheless, several conclusions as to the role of *p53* at the beginning of mammalian development are justified.

The ability of ESCs to undergo self-renewal is unaffected by the presence or absence of *p53*, revealing that *p53* does not play a significant role in maintaining expression of genes required for pluripotency (Section 6). However, the ability of ESCs to maintain genomic stability during self-renewal is *p53*-dependent (Section 5), thus *p53*’s role as ‘guardian of genome’ begins with formation of pluripotent stem cells. In addition, *p53* facilitates differentiation of pluripotent stem cells and inhibits reprogramming of differentiated cells into pluripotent cells (Section 7). Therefore, although *p53* is not essential for cell differentiation during gastrulation, it does facilitate this event.

The DNA damage response in embryonic cells appears more complex, but justifiable conclusions are possible by analysis of critical differences in experimental conditions (Section 8.5.1). In the case of naïve ESCs, a G1 phase cell cycle arrest is absent and a transient G2 phase cell cycle arrest is *p53*-independent. Moreover, DNA damage induced apoptosis occurs at the same rate and to the same extent in *p53*-/- ESCs as in *p53*+/+ ESCs. However, depending on experimental conditions, initiation of apoptosis is accelerated by several hours in *p53*+/+ ESCs (Section 8.5.2). When differences in experimental conditions are taken into account, the same conclusions could apply to 2iESCs, as well (Section 8.4 and Section 8.5.3). Results with EpiSCs have yet to be reported.

Given that fact that *p53* is neither essential for ESC renewal nor differentiation, which is evident from the facts that embryos lacking *p53* genes continue to gastrulate and develop, and that newborns thrive until succumbing to tumors, then how is the genome protected prior to gastrulation? The answer is that *p53* is neither required for cell cycle arrest nor apoptosis in either naïve or ground-state ESCs. Therefore, regardless of the cellular stress and regardless of the genomic defect, totipotent and pluripotent stem cells can undergo apoptosis prior to implantation. Thus, *p53*-independent apoptosis provides a mechanism for promoting implantation of healthy embryos.

The biological significance of what is learned through characterization of ESCs in vitro depends entirely on their ability to model events demonstrated in vivo. The lessons learned from studies on the DNA damage response in both 2iESCs and naïve ESCs demonstrate the challenges involved.

## Figures and Tables

**Figure 1 genes-12-01675-f001:**
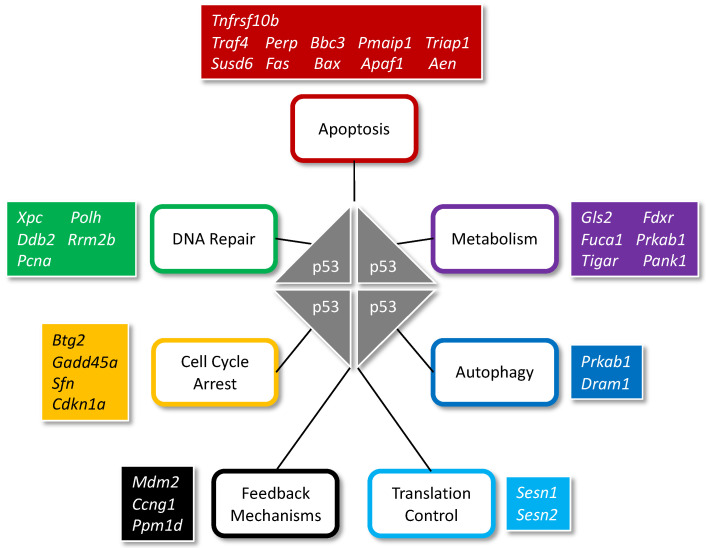
**High confidence *p53* target genes and the functions they facilitate**. *p53* binds to DNA as a homotetramer. Examples of proteins encoded by *p53* target genes function in multiple processes include, but are not limited to, cell cycle arrest, DNA repair, apoptosis, metabolism, autophagy, translation control and feedback mechanisms. Adapted from [32] with permission from Oncogene journal, 2017. Symbols are from the HUGO gene nomenclature committee (https://www.genenames.org/ accessed on 20 October 2021).

**Figure 2 genes-12-01675-f002:**
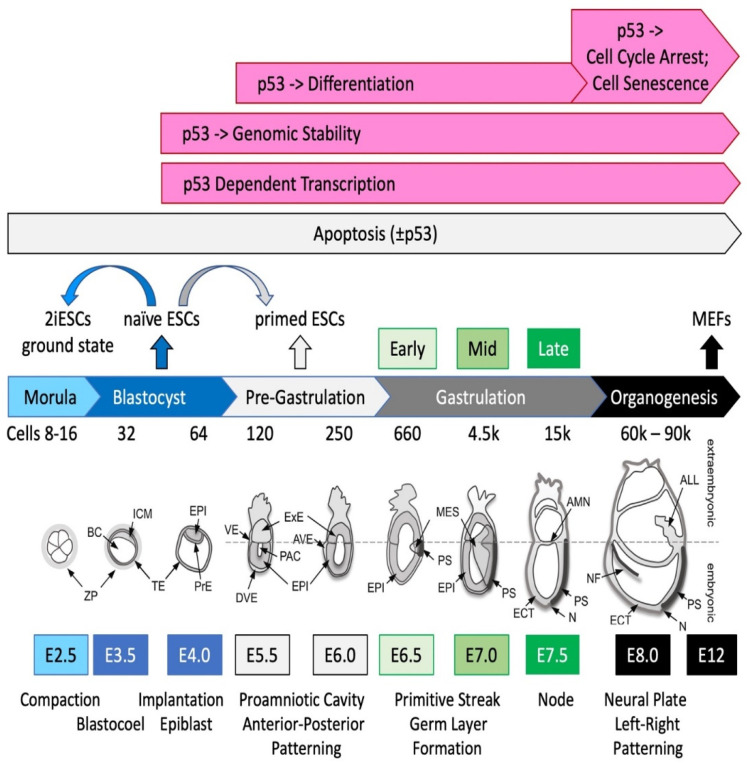
**Preimplantation and early post****-implantation mouse development**. The *p53* transcription factor and tumor suppressor protein is not essential for mammalian embryonic development, although it does facilitate several important events beginning with maintaining genomic stability by preventing polyploidy and aneuploidy during self-renewal of pluripotent embryonic stem cells (ESCs). Self-renewal refers to the ability of pluripotent stem cells to undergo an indefinite number of cell divisions without losing their ability to differentiate into cells derived from all three primary germ layers. In ESCs, *p53* facilitates differentiation during gastrulation, although it is not required for either transient cell cycle arrest or induction of apoptosis in response to DNA damage. In mouse embryonic fibroblasts (MEFs), DNA damage induces either *p53*-facilitated transient cell cycle arrest or cell senescence. The range or average number of cells for the entire conceptus (embryonic days post-coitum E0.5-E4.5) or the epiblast and the germ layers (E5.5-E8.0), and the key morphogenetic events at each age are indicated. Abbreviations: ALL (allantois), AMN (amnion), AVE (anterior visceral endoderm), BC (blastocyst cavity), DVE (distal visceral endoderm), ECT (ectoderm), EPI (epiblast), ExE (extraembryonic ectoderm), ICM (inner cell mass), MES (mesoderm), N (node), NF (neural fold), PAC (proamniotic cavity), PrE (primitive endoderm), PS (primitive streak), TE (trophectoderm), VE (visceral endoderm), ZP (zona pellucida). Adapted from [36] with permission from Placenta journal, 2004.

**Figure 3 genes-12-01675-f003:**
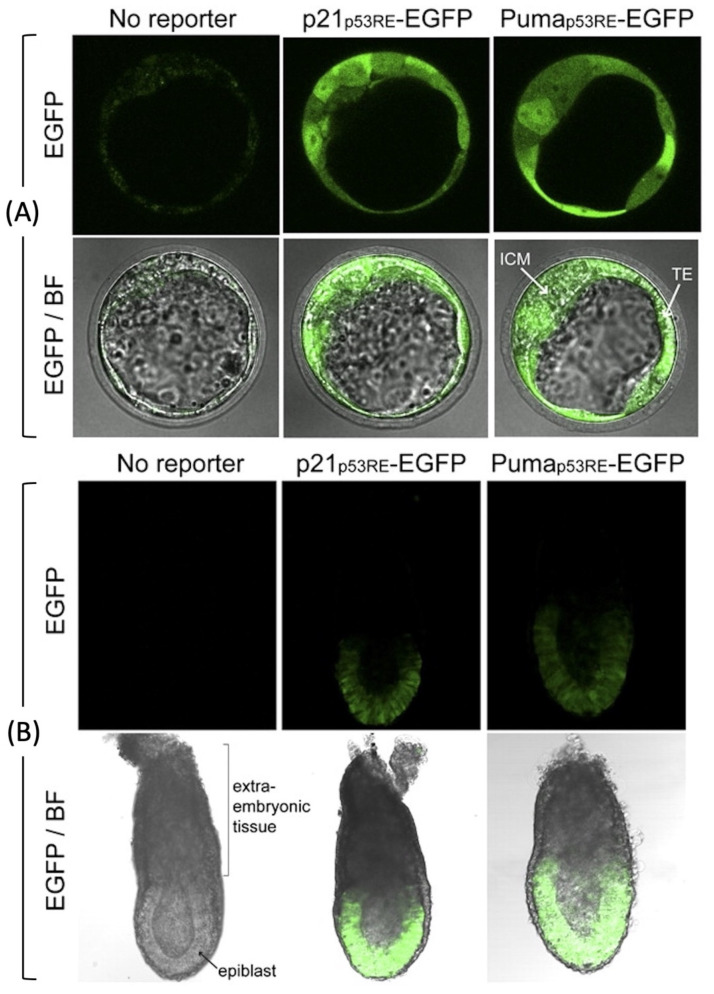
***p53* transcriptionally activity at the beginning of mouse development.***p53* activity was assayed in early stage embryos isolated from mice homozygous for reporter genes expressing enhanced green fluorescence protein (EGFP) from either the *p21* gene’s *p53* response element or the Puma gene’s *p53* response element and examined immediately by confocal microscopy, adapted from [47], with the permission from Proc. Natl. Acad. Sci. USA, 2012. (**A**) At embryonic day E3.5, EGFP fluorescence was detected in all blastocyst cells, including the inner cell mass (ICM), and trophectoderm (TE). The large blastocoel cavity identifies these examples as late-stage blastocysts. (**B**) EGFP fluorescence was also detected in E6.5 post-implantation reporter embryos but only in the epiblasts and not in the extraembryonic tissues. BF indicates bright field microscopy.

**Figure 4 genes-12-01675-f004:**
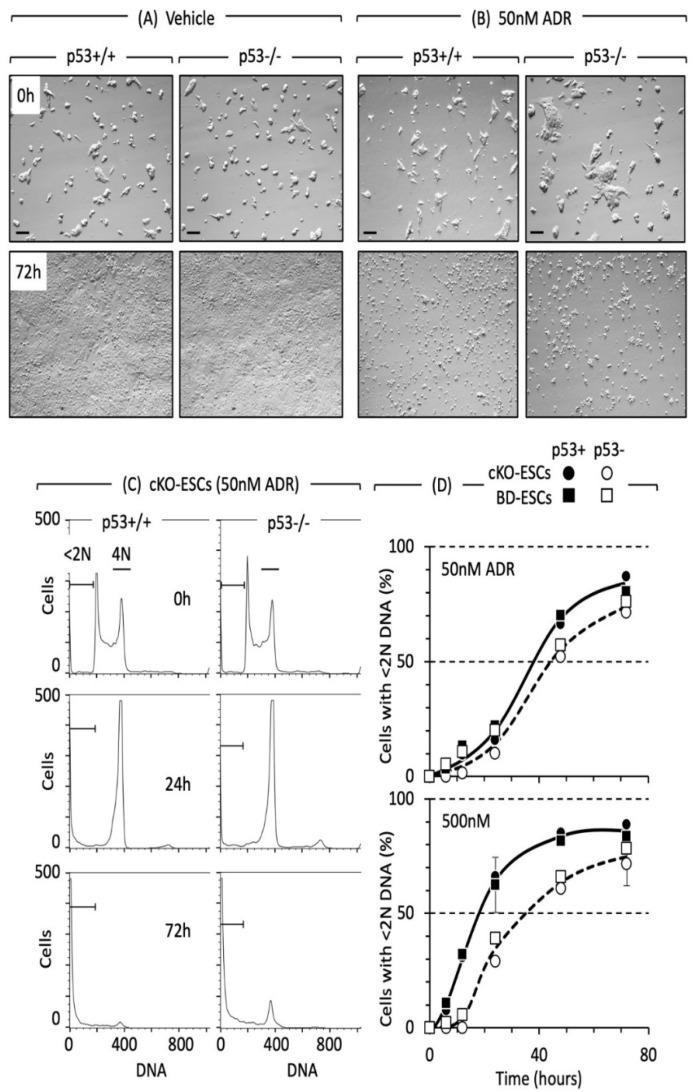
Cell cycle arrest and apoptosis in naïve ESCs are not dependent on *p53*. Low concentrations of Adriamycin/doxorubicin (ADR) were equally toxic to both *p53*+/+ and *p53*-/- mouse naïve ESCs. Blastocyst derived (BD) ESCs were seeded at 14,000 cells/cm^2^ and then cultured with either vehicle (**A**) or 50 nM ADR (**B**). Death was confirmed by combining attached and unattached cells and then staining with trypan blue. Results with conditional knockout (cKO) ESCs were indistinguishable from those with BD-ESCs. Figures are adapted from figures S1 (from [126]) and 1 of [126], with the permission from Stem Cells, 2020. (**C**) 50 nM ADR triggered the G2-checkpoint within 24 h and apoptosis within 72 h in both *p53*+/+ and *p53*-/- cKO-ESCs. Similar results were obtained with BD-ESCs. Attached and unattached cells were combined, their DNA stained with propidium iodide, and then fractionated according to DNA content using fluorescence activated cell sorting. Cells with <2N DNA content (apoptotic cells) and cells with 4N DNA content (G2/M phase cells) are indicated. (**D**) Cells with <2N DNA content were quantified as a function of time cultured with ADR. These data were normalized to 0% at zero hours. Error bars indicate ±SEM. Solid symbols are *p53*+/+ cells. Open symbols are *p53*-/- cells. cKO-ESCs are circles. BD-ESCs are squares. Panels C and D are examples from figures 2 and S2 in [126].

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
