# Peer review of "Developmental Acquisition of *p53* Functions"

_genes, 2021, doi:10.3390/genes12111675_

Round 1
Reviewer 1 Report
The manuscript summarizes and reconciles different data on p53-regulated events during development. The manuscript is very-well written and contains valuable information presented in a clear manner. In addition to literature review, the Authors extensively address previous contradictory results by providing detailed explanations. Furthermore, at the end of each section the Authors provide valuable take-home lessons. The manuscript is prepared carefully and all the necessary aspects are provided and discussed in a logical and orderly manner. Graphic abstracts and Figures are clear and exhaustive.
Only one minor edit/typo:
- Line 325, add “tail” or other subject: "Long non-coding RNAs (LncRNAs), >200 nucleotides with poly-adenylated tail but…"
Author Response
We have gone through the manuscript thoroughly. We have edited the manuscript for clarity, brevity, and consistency in presentation. We have addressed each point raised by the reviewers in the rebuttal letter as follow:
Reviewer #1
Comment: Line 325, add “tail” or other subject: "Long non-coding RNAs (LncRNAs), >200 nucleotides with poly-adenylated tail but…"
Response: We have added “tail”. Please see line # 312.
Reviewer #2
Major issues:
Comment: The section 1 “Current consensus on p53” is very compressed and choppy. Does it make sense to make separate paragraphs for the sake of two sentences? If the article mainly aims at a role of p53 in embryogenesis, a short introduction with a general characteristic of p53 without excessive structuring would be sufficient.
Response: We have reformatted the section 1 to keep a short introduction of p53. We moved remaining part to sections 4.2 (please see lines # 205-217).
Comment: The section 8.1.” Physiological States of Pluripotent Embryonic Stem Cells” contains a general information about ESC. It would be logical to place this section before the Section 6 “Maintaining the Pluripotent State” or merge both sections.
Response: We think the section 8.1.” Physiological States of Pluripotent Embryonic Stem Cells” is at its right place because in the following sections we have differentiated DNA damage response (cell cycle arrest and apoptosis) in different physiological states of ESCs.
Comment: Line 696: “Unregulated expression of p53 is not lethal in mice until after the blastocyst has implanted (E4.5) and gastrulation has begun (E6.25-E7.5)”. It has been shown that loss of Mdm2 which leads to deregulated accumulation of p53 in mouse embryo results in embryonic lethality at very early stage (3.5 dpc) (Chavez-Reyes et al., “Switching mechanisms of cell death in mdm2- and mdm4-null mice by deletion of p53 downstream targets”, Cancer Research, 2003).
Response:
Thank you very much for bringing the study by Chavez-Reyes et al., 2003 in our notice. Chavez-Reyes et al., 2003 stated initiation of apoptosis begins at 3.5dpc not the lethality. In this study authors performed tunnel assay in 3.5dpc isolated embryos which detects DNA damage that doesn’t confirm apoptosis or lethality of embryos. We have cited this paper our manuscript now. Please see lines # 239-245.
Comment: The section 10 “p53 and Cancer Stem Cells” is quite chaotic. In my opinion, the field that is covered by hundreds of publications is beyond the scope of this review (the role of p53 in development and embryonic cells). The fragmentary data make rather confusing impressions, especially the attempt to discuss aberrant activity of the hot-spot p53 mutant R175H (R172H in mice).
Response: We have removed the section 10 “p53 and Cancer Stem Cells”.
Minor issues
Comment: Line 102 unformatted reference
Response: We have corrected the reference format. Please see line # 36.
Comment: Line 114 The senescence is called „temporary cell cycle arrest”, although by definition senescence is an irreversible block of proliferation (in physiological conditions).
Response: Thank you very much for noticing this. We have corrected the sentence now avoiding confusion. Please see line # 686.
Comment: Line 344 The word “cells” is probably missing
Response: We have added the missing word “cells”. Please see line # 330.
Reviewer 2 Report
The review “Developmental Acquisition of p53 Functions“ written by Jaiswal, Raj and DePamphilis, provides a broad look at the role of p53 tumor suppressor in development and stress response during early embryogenesis. It encompasses multiple evidence from numerous publications and highlights an important difference between p53 functions in pluripotent and differentiated cells. Apart from the main topic, this review perfectly illustrates the problem of reproducibility and shows with real examples how differences in experimental conditions may lead to opposite conclusions. The manuscript is well structured, mostly clear and well written, but there are several issues that should be improved.
Major issues:
The section 1 “Current consensus on p53” is very compressed and choppy. Does it make sense to make separate paragraphs for the sake of two sentences? If the article mainly aims at a role of p53 in embryogenesis, a short introduction with a general characteristic of p53 without excessive structuring would be sufficient.
The section 8.1.” Physiological States of Pluripotent Embryonic Stem Cells” contains a general information about ESC. It would be logical to place this section before the Section 6 “Maintaining the Pluripotent State” or merge both sections.
Line 696: “Unregulated expression of p53 is not lethal in mice until after the blastocyst has im-696 planted (E4.5) and gastrulation has begun (E6.25-E7.5)”. It has been shown that loss of Mdm2 which leads to deregulated accumulation of p53 in mouse embryo results in embryonic lethality at very early stage (3.5 dpc) (Chavez-Reyes et al., “Switching mechanisms of cell death in mdm2- and mdm4-null mice by deletion of p53 downstream targets”, Cancer Research, 2003)
The section 10 “p53 and Cancer Stem Cells” is quite chaotic. In my opinion, the field that is covered by hundreds of publications is beyond the scope of this review (the role of p53 in development and embryonic cells). The fragmentary data make rather confusing impressions, especially the attempt to discuss aberrant activity of the hot-spot p53 mutant R175H (R172H in mice).
Minor issues
Line 102 unformatted reference
Line 114 The senescence is called „temporary cell cycle arrest”, although by definition senescence is an irreversible block of proliferation (in physiological conditions).
Line 344 The word “cells” is probably missing
Author Response

(The authors gave the same response as above.)
